# Neuronal and Astrocytic Differentiation from Sanfilippo C Syndrome iPSCs for Disease Modeling and Drug Development

**DOI:** 10.3390/jcm9030644

**Published:** 2020-02-28

**Authors:** Noelia Benetó, Monica Cozar, Laura Castilla-Vallmanya, Oskar G. Zetterdahl, Madalina Sacultanu, Eulalia Segur-Bailach, María García-Morant, Antonia Ribes, Henrik Ahlenius, Daniel Grinberg, Lluïsa Vilageliu, Isaac Canals

**Affiliations:** 1Department Genetics, Microbiology and Statistics, Faculty of Biology, University of Barcelona, CIBERER, IBUB, IRSJD, E-08028 Barcelona, Spain; 2Stem Cells, Aging and Neurodegeneration Group, Department of Clinical Sciences, Neurology, Lund Stem Cell Center, Lund University, SE-22184 Lund, Sweden; 3Department of Biochemistry and Molecular Genetics, Hospital Clinic de Barcelona, CIBERER, IDIBAPS, E-08028 Barcelona, Spain

**Keywords:** sanfilippo syndrome, mucopolysaccharidosis III, lysosomal storage disorders, induced pluripotent stem cells, neuronal differentiation, astrocyte differentiation, transcription factor-based differentiation, lysosomes, siRNAs, substrate reduction therapy

## Abstract

Sanfilippo syndrome type C (mucopolysaccharidosis IIIC) is an early-onset neurodegenerative lysosomal storage disorder, which is currently untreatable. The vast majority of studies focusing on disease mechanisms of Sanfilippo syndrome were performed on non-neural cells or mouse models, which present obvious limitations. Induced pluripotent stem cells (iPSCs) are an efficient way to model human diseases in vitro. Recently developed transcription factor-based differentiation protocols allow fast and efficient conversion of iPSCs into the cell type of interest. By applying these protocols, we have generated new neuronal and astrocytic models of Sanfilippo syndrome using our previously established disease iPSC lines. Moreover, our neuronal model exhibits disease-specific molecular phenotypes, such as increase in lysosomes and heparan sulfate. Lastly, we tested an experimental, siRNA-based treatment previously shown to be successful in patients’ fibroblasts and demonstrated its lack of efficacy in neurons. Our findings highlight the need to use relevant human cellular models to test therapeutic interventions and shows the applicability of our neuronal and astrocytic models of Sanfilippo syndrome for future studies on disease mechanisms and drug development.

## 1. Introduction

Lysosomal storage disorders (LSDs) are a group of rare inherited metabolic diseases caused by deficiencies in lysosomal enzymes leading to impaired recycling of macromolecules and alteration of the endolysosomal system [1]. Mucopolysaccharidoses (MPS), one type of LSD, arise from mutations in the genes responsible for the degradation of glycosaminoglycans (GAGs), which accumulate within the lysosomes. Among MPS, Sanfilippo syndrome (also known as mucopolysaccharidosis III or MPS III) is the most frequent form and is characterized by the incomplete degradation of one specific GAG known as heparan sulfate (HS). At the cellular level, partially degraded HS accumulates inside the lysosomes of several organs and tissues [2,3,4]. Four different Sanfilippo syndrome subtypes are recognized depending on the mutated gene and consequent enzyme deficiency: type A (OMIM#252900), type B (OMIM#252920), type C (OMIM#252930), and type D (OMIM#252940). Patients from all four subtypes show similar clinical symptomatology, mainly characterized by an early-onset severe and progressive neurodegeneration accompanied by mild somatic symptoms [2,3,4]. The incidence of all Sanfilippo syndrome subtypes is around 1 in 70,000 live births, with a prevalence of 1 to 9 in 1,000,000 people depending on the studied population. Prevalence of the different subtypes vary between populations (i.e., subtype A is more frequent in the north of Europe while subtype B is more frequent in southern Europe [5]). Sanfilippo syndrome type C is caused by a deficiency in an enzyme located in the lysosomal membrane, heparan-alpha-glucosaminide N-acetyltransferase (HGSNAT, EC 2.3.1.78) [6], which is encoded by the *HGSNAT* gene. This gene is in the pericentromeric region of chromosome 8 (8p11.2–8p11.1) and has 18 exons [7,8]. The HGSNAT protein has 635 amino acids and 11 transmembrane domains [9]. Sanfilippo syndrome type C presents a prevalence of 1 in 1,500,000 live births, accounting for approximately 4% of all Sanfilippo syndrome cases worldwide [3]. First neurological symptoms appear at an early age (commonly within 3 to 7 years of age) and patient life expectancy spans from 10 to 30 years [3].

To date, there is no treatment for the neurological symptoms of Sanfilippo syndrome, and management of these patients consists of palliative measures. For non-neurological LSDs, enzyme replacement therapy has been proven to be the most successful strategy [10]; however, the blood–brain barrier limits availability of the enzyme in the brain and intrathecal administration, besides being a very invasive strategy, did not promote neurocognitive benefits in most Sanfilippo patients in a recent clinical trial [11]. Similarly, therapies using hematopoietic stem cell transplantation before disease onset, although useful for treating somatic symptoms, are not effective to prevent neurodegeneration in patients [12]. Alternatively, the use of pharmacological chaperones to improve the correct folding and stability of the defective protein has been approved for some LSDs [13]. For Sanfilippo syndrome type C, promising results were shown using glucosamine in patients’ fibroblasts [14], but its efficiency in brain cells and its ability to cross the blood–brain barrier remains to be assessed. Gene therapy is an optimal therapeutic option for LSDs since it has been proposed that increases around 10% in enzymatic activity are sufficient to produce clinical benefits in patients [10]. In the case of Sanfilippo syndrome types A and B, two clinical trials based on intracerebral injection of adeno-associated virus (AAV) showed some neurological improvements in patients [15,16]. However, it is important to note that successful gene therapy for lysosomal enzymes relies on the ability of transduced cells to share the correct lysosomal enzyme with non-transduced neighboring cells through 6-mannose phosphate receptors [17]. Considering that HGSNAT is a lysosomal transmembrane protein that does not shuttle through the 6-mannose phosphate pathway, Sanfilippo C syndrome might not be the best candidate for this therapeutic strategy. Nonetheless, some promising results have been obtained in a mouse model using a novel AAV with a modified capsid [18]. Another interesting therapeutic approach for LSDs is substrate reduction therapy (SRT) to decrease the synthesis of the molecule that cannot be correctly degraded. For Sanfilippo syndrome, rhodamine B and genistein have shown good results in fibroblasts or animal models [19,20,21], however, those results did not translate in clear neurological benefits for patients [22]. A different SRT approach consists in the use of RNA interference (RNAi) to inhibit genes responsible for GAG synthesis. Patients’ fibroblasts treated with siRNAs or shRNAs against two genes involved in HS synthesis showed a clear reduction in GAG production [23,24,25] and HS storage [25]. However, given the neurological symptoms seen in patients, it is crucial to study SRT in relevant human neural cells.

For many years, human culture systems were limited to the use of immortalized cell lines with genetic and epigenetic aberrations as well as unstable karyotypes or primary cells from patients, which are very difficult to obtain [26]. Moreover, patient cells are usually derived from postmortem material, which represents the end stage of the disease and does not allow studies on early disease-related alterations. Fibroblasts are often used as human cellular models in LSDs, but there are significant differences between fibroblasts and neural cell types. All these aspects accentuate the importance of generating new relevant cell models to investigate the underlying mechanisms of disease. The discovery of strategies to reprogram somatic cells back to pluripotency [27] has created several opportunities for generating in vitro models of rare monogenic diseases of the nervous system.

Due to the lack of alternative sources, induced pluripotent stem cell (iPSC)-derived neurons and astrocytes are particularly valuable for studies of human disease mechanisms. In the last years, several differentiation protocols to differentiate iPSCs into neurons have been described [28]. Nevertheless, neurons are not the only neural cell type involved in neurological disorders. Research in the last 20 years has emphasized the role of glial cells, especially astrocytes, in the regulation of brain functionality and homeostasis [29]. For that reason, several differentiation protocols to generate astrocytes from iPSCs have been developed [28]. However, these protocols are usually time-consuming and technically challenging and obtained cells are not always well characterized. Transcription factor-based strategies to accelerate differentiation of iPSCs into pure populations of specific cell types have been recently described [30]. These protocols represent a very useful tool considering the need for large-scale production of neurons and astrocytes with high purity for clinical applications and drug screening. Initially, this strategy was shown to drive iPSC differentiation towards excitatory cortical induced neurons (iNs) by lentiviral overexpression of Ngn2 [31]. With this approach, efficient conversion to a pure population of excitatory neurons with functional synapses was achieved in two weeks. Regarding glial differentiation, functional and mature induced astrocytes (iAs) from iPSCs were obtained in three weeks after lentiviral overexpression of two gliogenic transcription factors, Nfib and Sox9 [32]. These transcription factor-based protocols are valuable tools to rapidly and efficiently generate in vitro models to investigate in depth the role of neurons and astrocytes in neurodegenerative LSDs, as well as to assess and optimize potential therapies through drug and toxicity screening.

The recent development of the CRISPR/Cas9 system [33], an RNA-based genome-editing tool, enables efficient site-specific genome editing to generate isogenic cell lines from iPSCs, either introducing specific mutations in a healthy line or correcting mutations in patient-derived lines. The main advantage of having isogenic lines is to avoid the possibility of detecting non-disease-related phenotypes arising from differences in the genetic background of patients and controls, one of the main drawbacks of iPSC-based studies using several lines. By combining iPSCs with CRISPR/Cas9 genome editing, it is possible to generate models where the only genetic difference between cell lines is the disease-causative mutation. Importantly, after CRISPR/Cas9 editing, the capacity of iPSCs to rapidly proliferate and differentiate remains unaffected due to the high target specificity of this technique [34].

Two previous studies have been performed using Sanfilippo patient-derived iPSCs and differentiation towards the neural lineage [35,36]. In the first study on Sanfilippo B iPSC-derived neurons differentiated for five weeks, authors found bigger intracellular vesicles positive for lysosomal associated membrane protein 1 (LAMP1) together with increased and abnormal Golgi complexes. Moreover, gene expression profiles suggested alterations of extracellular matrix constituents as well as cell–matrix interactions during differentiation [35]. However, glial cells were not investigated in this work and their role in disease mechanisms remained unknown. In a second study, Sanfilippo C iPSC-derived neural cultures showed also bigger vesicles positive for LAMP1 and accumulation of GAGs after nine weeks of differentiation [36]. Interestingly, network activity and connectivity were impaired, thus suggesting that lysosomal alterations were leading to functional impairment. Nevertheless, specific astrocyte-phenotypes were again not investigated, keeping unanswered what the exact role is of astrocytes in disease development. 

We have previously generated two HGSNAT-mutated iPSC lines [37] through the use of CRISPR/Cas9 and one iPSC line from a Sanfilippo syndrome type C patient’s fibroblasts [36]. Here we combined these three iPSC lines with optimized protocols to obtain neurons [31] and astrocytes [32], creating novel and relevant disease models that recapitulate major Sanfilippo syndrome hallmarks. We then assayed an siRNA-based SRT strategy that was successful in treating patient fibroblasts [25] and showed that this strategy is not effective in neural cells, highlighting the importance of using disease-relevant cells in studies of disease mechanisms and drug screening.

## 2. Experimental Section

### 2.1. Human iPSCs

Four previously generated iPSC lines were used: one healthy control (WT) and a patient-derived line (SFC6) [36], and two isogenic mutant lines, HGSNAT1 and HGSNAT2, generated from the WT iPSC line with CRISPR/Cas9 genome editing [37].

All iPSCs were maintained in feeder-free conditions using mTeSR™ Plus medium (STEMCELL Technologies, Grenoble, France) with 0.5% Penicillin Streptomycin (P/S, Thermo Fisher Scientific, Waltham, MA USA) on Matrigel (Corning, Corning, NY USA)-treated plates and maintained at 37 °C in humidified air with 5% CO_2_, with medium changed every 2–3 days. Cells were passaged with StemPro Accutase Cell Dissociation Reagent (Accutase, Thermo Fisher Scientific) every 3–4 days, when reaching approximately 80% confluency and plating at 2 × 10^4^ cells/cm^2^. To improve survival rate, either 2 µM Thiazovivin (TZV, STEMCELL Technologies) or 10 µM Rock Inhibitor (RI, STEMCELL Technologies) was added to the medium for 24 h after plating.

References for all products can be found in Appendix A.

### 2.2. Lentiviral Production

Lentiviral vectors used were M2-rtTA (rtTA, reverse tetracycline-controlled transactivator, Addgene, Watertown, MA USA), tet-O-Ngn2-puro (Ngn2, Addgene), tetO-Sox9-Puro (Sox9, [32]), and tetO-Nfib-Hygro (Nfib, [32]).

Ngn2, Sox9, Nfib, and rtTA lentiviruses were produced in HEK 293T cells as previously described [32]. Briefly, cells were cotransfected with lentivectors and the packaging plasmids pMD2.G (Addgene), pRSV-Rev (Addgene), and pMDLg/pRRE (Addgene) using 2.5 M CaCl_2_. For two T175 flasks, 22 µg of pMD2.G, 15 µg of pRSV-Rev, 30 µg of PMDLg/pRRE, and 75 µg of the desired lentivector plasmids were transfected. The day after, medium was changed and 24 h later, viruses were harvested and pelleted by centrifugation (20,000× *g* for 2 h at 4 °C), supernatant was aspirated, and 100 µL of Dulbecco’s modified Eagle’s medium (DMEM, Merck KGaA, Darmstadt, Germany) added to the virus pellet without resuspending. The day after, viruses were resuspended, aliquoted, and stored at −80 °C.

References for all products in Section 2.2 are in Appendix A.

### 2.3. Generation of Induced Neurons and Astrocytes from iPSCs

Both neural induction [31] and astrocyte induction [32] were carried out as previously described with minor modifications. Protocol schemes are shown in Figure 1.

Briefly, on day 2, human iPSCs at approximately 80% confluency were dissociated with Accutase, and 7 × 10^5^ cells were replated in Matrigel-coated six-well plates using mTeSR™ Plus medium with 10 μM RI. One day later (day 1), medium was replaced by fresh mTeSR™ Plus medium, and 1 μL of each virus (rtTA and Ngn2 for iNs; rtTA, Nfib, and Sox9 for iAs) was added to each well. On day 0, medium was replaced with fresh mTeSR™ Plus medium containing 2.5 μg/mL doxycycline (Dox, Thermo Fisher Scientific), which was kept in the medium throughout the experiments.

For neuronal induction, from day +1, medium was changed daily using N2B27 medium (1:1 DMEM/F12 (Thermo Fisher Scientific) and Neurobasal medium (Thermo Fisher Scientific), 1% (50×) B-27™ supplement (Thermo Fisher Scientific), 0.5% (100×) N-2 supplement (Gibco™, Thermo Fisher Scientific), 1% GlutaMAX (Thermo Fisher Scientific), and 1% P/S) with 48 h of 1.25 μg/mL puromycin (Thermo Fisher Scientific) selection. On day +6, cells were dissociated with Accutase over 20 min, and passed through 40 µm strainers (Corning) to eliminate cell aggregates. Then, 5 × 10^5^ cells were replated in Matrigel-treated six-well plates for real time quantitative polymerase chain reaction (RT-qPCR) and ultra-performance liquid chromatography–tandem mass spectrometer (UPLC-MS/MS) heparan sulfate (HS) quantity measurement experiments, and 7 × 10^4^ cells were replated in Matrigel-treated 13 mm coverslips for immunocytochemistry experiments.

For astrocyte induction, on days +1 and +2, cells were cultured in Expansion medium (DMEM/F-12, 10% fetal bovine serum (FBS, Thermo Fisher Scientific), 1% (100×) N-2 supplement, and 1% GlutaMAX). On day +3, 75% of Expansion medium was combined with 25% of FGF medium (Neurobasal, 2% (50×) B-27™ supplement, 1% non-essential amino acids (NEAA, Thermo Fisher Scientific), 1% GlutaMAX, and 1% FBS, 8 ng/mL fetal growth factor (FGF, Peprotech, London, UK), 5 ng/mL ciliary neurotrophic factor (CNTF, Peprotech), and 10 ng/mL bone morphogenetic protein 4 (BMP4, Peprotech)). On day +4, 50% of Expansion medium was combined with 50% of FGF medium. On day +5, 25% of Expansion medium was combined with 75% of FGF medium. Selection of transduced cells was carried out on days 1–2 adding 1.25 μg/mL puromycin and on days 1–5 adding hygromycin (200 μg/mL) (Thermo Fisher Scientific). On day +6, cells were dissociated with Accutase until a single cell suspension was achieved. Then, 5 × 10^5^ cells were replated in Matrigel-treated six-well plates for RT-qPCR experiments using FGF medium alone. For immunocytochemistry, 7 × 10^4^ cells were replated in Matrigel-treated 13 mm coverslips using FGF medium alone.

References for all products in Section 2.3 are in Appendix A.

### 2.4. siRNA Transfection

Neurons and astrocytes grown in six-well plates were transfected on day +7 using 2 μL per well of Lipofectamine RNAiMAX transfection agent (Thermo Fisher Scientific). One siRNA Silencer^®^ Select against *EXTL2* gene (siRNA-4899, Assay ID: si4899, Thermo Fisher Scientific) and one negative control siRNA (siRNA-C, Thermo Fisher Scientific) were used at a final concentration of 30 nM and one well was treated with Lipofectamine RNAiMAX alone (No-siRNA) as a control.

References for all products in Section 2.4 are in Appendix A.

### 2.5. Immunofluorescence Staining

Neurons or astrocytes were fixed on day +10 with 4% paraformaldehyde (PFA, VWR, Llinars del Vallés, Spain) for 15 min, washed with tris-buffered saline (TBS, Thermo Fisher Scientific), blocked and permeabilized with TBS containing 0.1% Triton-X 100 (VWR) and 5% normal donkey serum (Merck KGaA, TBS++) for 2 h at room temperature. Primary antibodies were incubated overnight at 4 °C. The day after, cells were washed twice for 5’ with TBS, 5’ with TBS++ and secondary antibodies and 0.5 μg/mL 4’,6-diamidino-2-fenilindol (DAPI, Thermo Fisher Scientific) or 1 μg/mL Hoechst 33342 (Thermo Fisher Scientific) were incubated 2 h at room temperature in TBS++. After washing 5’ with TBS for three times and once with ddH_2_O, coverslips were mounted on slides with Mowiol® 4-88 (Merck KGaA) mounting medium.

Images were acquired using ZEISS confocal microscope, and then all images were mounted using the Fiji-ImageJ software [38]. Analysis of images was performed with summation of 10 z-stacks separated by 0.36 µm. First, soma area was defined by Tuj1+ staining and, within this area, the lysosomal associated membrane protein 2 (LAMP2) area was measured establishing a threshold with Li algorithm. To obtain LAMP2/Tuj1 area, both areas obtained were divided for each z-stack summation. Mean intensity data were obtained directly measuring the mean intensity inside the area established by the LAMP2 threshold.

References for all products in Section 2.5 are in Appendix A.

References for antibodies used in Section 2.5 are in Appendix A.

### 2.6. RT-qPCR

RNA was extracted at day +10 using High Pure RNA Isolation Kit (Roche, Basel, Switzerland), and 2 µg of total RNA was used to synthesize cDNA using the High-Capacity cDNA Reverse Transcription kit (Thermo Fisher Scientific) and the RNase Inhibitor (Thermo Fisher Scientific), following manufacturer’s instructions. Real-time qPCR was performed in LightCycler 480 II (Roche) system with a LightCycler 480 Probes Master (Roche) and TaqMan Gene Expression Assays (Thermo Fisher Scientific). Quantification cycle (Cq) calculation was done using LightCycler 480 Software (release 1.5.0, Roche). To confirm reproducibility of RT-qPCR, the intra-assay precision was determined in three repeats within one LightCycler run. Intra-assay coefficient variation of all assays at our working conditions was below 1% and Cq standard deviation smaller than 0.3. In all experiments, GAPDH expression was stable and used as a normalizer.

References for all products in Section 2.6 are in Appendix A. References for TaqMan assays used in Section 2.6 are in Appendix A.

### 2.7. HS Quantity Measurement

For HS quantity measurement experiments (UPLC-MS/MS), cells were collected at day +10 using cell scrapers (Labclinics, Barcelona, Spain), centrifuged at 300× *g* for 5 min, resuspended on 70 uL of ddH_2_O, and subjected to a four-cycle freezing (−80 °C)/thawing (4 °C) protocol to lysate the cells. Protein concentration was measured using DC™ Protein Assay (Bio-Rad, Hercules, CA USA), based on the Lowry method (reference in Appendix A) and HS was quantified using an established protocol based on UPLC-MS/MS [39].

### 2.8. Data Analysis

Data are presented as the mean ± s.e.m. of three independent experiments. Statistical analyses were performed using an appropriate test in Prism software. For cell-specific markers RT–qPCR, ratio paired t-test was performed. For area and intensity measurements of LAMP2, as well as for UPLC-MS/MS HS measurements, we used ordinary one-way ANOVA corrected by Dunnett post hoc test, comparing WT-iNs with the other groups. In the case of *EXTL2* mRNA expression RT-qPCR after siRNA treatment, an ordinary one-way ANOVA corrected by Tukey post hoc test was used. Significance was set at *p* < 0.05 for all experiments.

## 3. Results

### 3.1. Generation of iNs and iAs to Model Sanfilippo C Syndrome

To develop new neuronal and astrocytic models for Sanfilippo C syndrome, we differentiated iPSC lines into induced neurons (iNs) and induced astrocytes (iAs) using lentiviral overexpression of Ngn2 (iNs) or Sox9 and Nfib (iAs) as previously described [31,32]. We used one healthy control iPSC line (WT) and three disease iPSC lines, two generated from the WT line using CRISPR/Cas9 (HGSNAT1 and HGSNAT2) and one patient-derived iPSC line (SFC6). All mutant lines carried mutations in both alleles of the *HGSNAT* gene leading to decreased enzymatic activity [36,37].

To confirm differentiation, we examined expression of cell-type-specific markers using qPCR and immunocytochemistry. At day 10 after induction, pluripotency-related genes NANOG and POU51F1 (Figure 2A,B, respectively) were pronouncedly downregulated. On the other hand, neuronal-specific genes TUBB3, SYP, and MAP2 (Figure 2C–E) were highly upregulated in iNs compared to iPSCs, confirming acquisition of neuronal fate. Similarly, an evident increase in expression of astrocyte-specific genes GFAP, S100B, and ALDH1L1 (Figure 2F–H, respectively) was observed after iPSC differentiation into iAs.

To further confirm differentiation towards neurons and astrocytes, we analyzed expression of cell-type-specific markers by immunocytochemistry 10 days after induction. For neurons, Tuj1 and MAP2 were detected in a majority of iNs while for astrocytes, S100B and VIM were found in most iAs. Results were the same for iNs and iAs differentiated from all iPSC lines (Figure 3), indicating similar and high potential to rapidly generate pure populations of neurons and astrocytes with transcription factor-based protocols for all iPSC lines.

### 3.2. iNs and iAs Recapitulate Major Sanfilippo C Phenotypes

Next, we sought to evaluate whether iNs and iAs derived from disease lines recapitulated main Sanfilippo syndrome phenotypes and would represent good cellular models of the disease. For that purpose, we performed two different experiments: LAMP2 immunocytochemical analysis to detect alterations in the lysosomal content of disease cells and HS quantification through UPLC-MS/MS to compare HS amounts between disease and healthy cells.

LAMP2 has been extensively used as a marker for lysosomes. Using immunocytochemistry (Figure 4A), we found an increase of about 5% in the area occupied by LAMP2+ vesicles in the neuronal soma of HGSNAT2-iNs compared to WT-iNs, but not in the other lines (Figure 4B). When LAMP2 intensity was assessed, all disease lines showed a tendency to have increased levels compared to the WT (Figure 4C). Our results suggest that there is a slight increase in the lysosomal content of disease iNs compared to healthy iNs after only 10 days of in vitro differentiation from iPSCs.

As Sanfilippo syndrome patients accumulate HS and previous studies in fibroblasts confirmed this phenotype [25], we measured the amount of HS in iNs obtained from all iPSC lines with UPLC-MS/MS. We identified a tendency towards increased levels of HS in disease iNs compared to the healthy control, suggesting that iNs displayed one of the major cellular phenotypes associated with Sanfilippo syndrome (Figure 5).

### 3.3. Short-Term siRNA-Based SRT Is Not Efficient in Disease-Relevant Cells

One of the main applications of iPSCs and transcription factor-based protocols to induce differentiation is to test therapeutic approaches in relevant cell types that are difficult to obtain directly from patients. We wanted to evaluate an SRT based on siRNA that we have successfully applied on patients’ fibroblasts [25]. We selected an siRNA against the *EXTL2* gene, which codes for a protein involved in the first steps of HS-specific biosynthesis. This siRNA showed promising results in decreasing *EXTL2* expression and reducing GAG synthesis as well as HS storage in patient fibroblasts [25]. To test its therapeutic potential in iNs and iAs, we evaluated how mRNA levels and HS amounts were changed after short treatment with the siRNA.

Both iAs and iNs were treated independently with two siRNAs, one against *EXTL2* (siRNA-4899) and a control siRNA (siRNA-C). We observed a high reduction (around 75%) of *EXTL2* mRNA in both iNs and iAs differentiated from all disease iPSC lines compared with cells treated only with transfection reagent or a control siRNA (Figure 6). These results confirmed that siRNA-4899 was efficiently downregulating *EXTL2* mRNA in iNs and iAs as previously found in fibroblasts [25].

We next sought to elucidate whether the decrease in *EXTL2* mRNA levels would lead to reduced amounts of HS in treated iNs. To answer this question, we compared HS levels in iNs derived from the HGSNAT2 cell line treated with the siRNA-C or with the siRNA-4899. Although siRNA treatment reduced *EXTL2* mRNA levels by 75%, we could not detect a decrease in HS amounts compared to cells treated with siRNA-C. These results are in contrast with previous data obtained with patient fibroblasts in which HS levels were significantly decreased three days after treatment [25].

## 4. Discussion

In this study, we demonstrate that using protocols to produce iNs and iAs from iPSCs is an easy, efficient, and rapid way to generate relevant human cells for disease modeling of LSDs. For years, Sanfilippo syndrome cellular studies aiming to clarify disease mechanisms or to test therapeutic approaches have been performed on nonrelevant cells [25], on brain cells obtained with differentiation following developmental cues [35,36], or in murine models [40,41]. Although animal models can be useful, they do not accurately recapitulate human brain cells, which have a higher degree of morphological and functional complexity. Moreover, human cellular models may provide new, complementary, and useful information that might not be possible to obtain from animal models. Differentiation protocols dependent on sequential treatment with signaling molecules based on developmental cues are time-consuming and usually yield variable cell populations. This hampers reproducibility and comparability between different iPSC lines and performing high-throughput studies may be challenging. To bypass all these issues, we have in this study used novel transcription factor-based protocols to generate new neuronal and astrocytic cellular models for Sanfilippo syndrome type C that will facilitate studies on disease mechanisms and therapeutic approaches.

We have differentiated iNs and iAs from three different previously generated iPSC lines carrying mutations in the *HGSNAT* gene, responsible for Sanfilippo C syndrome. One of these lines was derived from Sanfilippo syndrome type C patient fibroblasts (SFC6) [36] and the other two were obtained by CRISPR/Cas9 gene editing from a healthy control iPSC line [37]. Importantly, the two engineered lines were isogenic to the WT healthy control used in this work, excluding the risk of detecting phenotypes arising from different genetic backgrounds instead of the disease-causative mutation, one of the main drawbacks of iPSC studies using lines from different donors. After 10 days of differentiation, cell identity was confirmed by expression of several cell-type-specific markers both at the mRNA and protein level. Major changes in differentiation efficiency were not detected, suggesting that Sanfilippo syndrome type C mutations do not hamper neuronal or astrocytic differentiation. This is not surprising considering that iNs and iAs protocols accelerate iPSC differentiation by overexpression of lineage-specific transcription factors, which could veil developmental impairments that might occur in patients. However, we have previously differentiated the SFC6 line together with the WT and another disease line using a protocol based on developmental cues and no significant differences in efficiency were found [36], suggesting that neuronal and astrocytic differentiation from iPSC is not affected by the presence of disease-causative mutations.

Importantly, we found that iNs displayed increased HS levels, one of the hallmarks of the disease at the cellular level, together with a slight enlargement in the lysosomal compartment of disease-iNs compared with WT-iNs based in the somatic area occupied by lysosomes and intensity of LAMP2+ vesicles. In a previous work, iPSC-derived neural cultures showed an increase of about 10% in LAMP2-positive vesicles in the soma of patient-derived lines after nine weeks of differentiation [36]. Here, iNs were differentiated for only 10 days, which could explain the lower degree of lysosomal enlargement. Altogether, our results suggest that an increased number and/or size of lysosomes are found in the neuronal soma of disease iNs, supporting the hypothesis that pathological phenotypes are starting to arise in this cellular model although longer times of differentiation might be needed to detect significant changes. In line with these results, HS also showed a clear tendency to accumulate in all Sanfilippo-derived iNs, recapitulating the main phenotype of the disease. This result suggests that transcription factor-based protocols were able not only to speed up the differentiation process but also to generate cells with earlier storage of HS. Here, disease iNs showed a 3.5-fold increase in HS when compared to a healthy control, while in a previous work, a 2-fold increase in GAGs was found only after nine weeks of differentiation [36], a difference that could be due to various factors. First, here we used a differentiation protocol that generates a pure population of excitatory neurons, while in the previous work iPSC-derived neural stem cells were differentiated towards a pan-neuronal phenotype mixed with astrocytes. Thus, excitatory neurons could accumulate higher amounts of HS or have a faster degree of accumulation. Second, here we measured HS with UPLC-MS/MS, while in the previous work we measured amounts of GAGs. HS is the specific GAG stored in Sanfilippo syndrome, but there are five different types of sulfated GAGs that could not be differentiated with the quantification of total GAGs. This issue could lead to an underestimation of specific HS differences. Nevertheless, our results confirm that iNs recapitulate major Sanfilippo syndrome phenotypes and will be a very valuable model in future studies on LSD mechanisms and drug screening.

It is essential to generate human cellular models with early alterations mimicking those occurring in Sanfilippo patients to have better and faster tools for disease modeling and drug screening studies. In previous works, disease-related alterations were detected at later stages of differentiation and differentiation protocols mimicking developmental cues were used [35,36], obtaining mixed populations of neurons and astrocytes. However, specific phenotypes in astrocytes were not investigated. Here, we show that Sanfilippo iPSC lines can be quickly and efficiently differentiated to pure populations of iNs and iAs, allowing to independently generate each cell type and investigate cell-specific phenotypes and meeting the requirements for scalable and high-throughput studies. This highlights the benefit of using transcription factor-based protocols in studies of disease mechanism and therapeutic approaches for LSDs with neurological affectation.

To support this idea, we tested an siRNA-based SRT previously assessed in patient fibroblasts [25] to evaluate its potential in relevant human cells. HS biosynthesis occurs in the Golgi compartment, where a linkage region in a core protein is formed as a starting point for GAG chain elongation. Synthesis of the linkage region is common for other GAGs, however, HS chain formation is specifically carried out by proteins of the EXT (EXT1 and EXT2) and EXTL (EXTL1, EXTL2, and EXTL3) families [42]. Considering that dominant mutations in *EXT1* and *EXT2* genes cause hereditary multiple exostoses [43] and the *EXTL1* gene is expressed at very low levels in brain cells [44], *EXTL2* and *EXTL3* represent the most promising candidates in this pathway for SRT. In Sanfilippo fibroblasts, an siRNA-mediated reduction of *EXTL2* and *EXTL3* mRNAs was shown to decrease amounts of GAGs and, specifically, HS after three days of treatment. Here, we decided to test the most efficient siRNA (targeting the *EXTL2* gene) in iNs and iAs derived from our three mutated lines. In both cell types, siRNA treatment led to a substantial reduction (up to 75%) of *EXTL2* mRNA levels, consistent with what was previously found in fibroblasts [25]. However, and in contrast to what occurred in fibroblasts, this reduction did not promote a decrease in HS levels of iNs, a difference that could be due to several causes. Fibroblasts are often used as human cellular models in LSDs, but there are significant metabolic differences between fibroblasts and neural cell types. Although fibroblasts accumulate undegraded materials, storage can be underestimated due to dilution by cell division [26], which can mitigate phenotypes that would otherwise be prominent in neuronal cells. Moreover, there are intrinsic differences in each cell type that might lead to changes in the capacity of cells to remove the already synthesized HS after treatment. In addition, some studies suggest that EXTL2 could act as a suppressor of HS biosynthesis in the liver of a mouse model [45] and liver cells of mice deficient in *EXTL2* produced significantly more HS during regeneration [46]. Future studies should better clarify the role of *EXTL2* in HS synthesis in neurons and astrocytes. At the moment, our results suggest that inhibition of *EXTL2* mRNA in iNs is not sufficient to reduce HS storage or that this inhibition should be maintained longer in order to see an effect on HS amounts in this cell type. Importantly, our results clearly demonstrate the importance of using relevant cell types for drug screening studies in order to test the efficacy of potential treatments in the target human cell.

It remains to be evaluated whether siRNAs against *EXTL3* reduce HS storage in iNs, considering that alternative functions have not been described for EXTL3. In addition, it will be interesting to examine LAMP2- and HS-related phenotypes in iAs, due to the important role of astrocytes in neurological disorders [29] and the potential of SRT targeting *EXTL2* and *EXTL3* in iAs. Finally, another important aspect that has to be further investigated is which class of HS is predominantly accumulated and the sulfation pattern of HS chains in each cell type as well as the presence of secondary aggregates of GAGs other than HS both in neurons and astrocytes. 

## 5. Conclusions

In this study, we have generated Sanfilippo C syndrome neurons and astrocytes that recapitulated major hallmarks of the disease using transcription factor-driven protocols that allow fast and efficient differentiation of iPSCs. Combining these two techniques will allow rapidly generating substantial new data to get insights on the basis of this neurodegenerative disease in patient relevant cell types, in contrast to commonly used nonrelevant cells like fibroblasts. Moreover, we provide evidence that these relevant cellular models provide a unique and important tool for drug screening after testing an siRNA-based SRT that proved to be successful on patient fibroblasts and failed in neurons. Our data highlights the importance of using appropriate relevant human cell types in future studies to avoid the limitations of using somatic cells to model neurodegenerative diseases. We strongly anticipate that iNs and iAs will be useful tools to rapidly evaluate therapeutic strategies in relevant human brain cell models of neurological disorders.

## Figures and Tables

**Figure 1 jcm-09-00644-f001:**
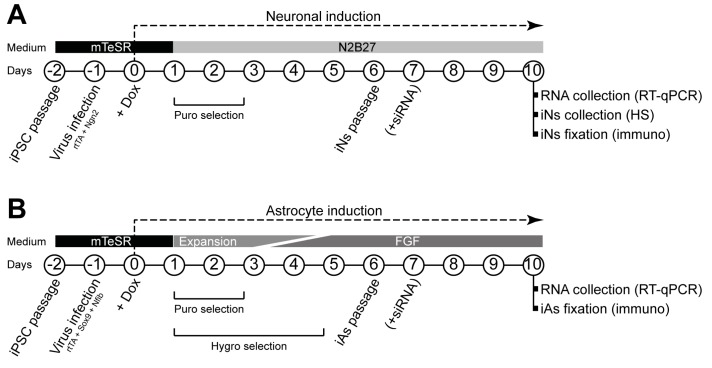
Schemes describing the protocols for induction of neurons (**A**) and astrocytes (**B**).

**Figure 2 jcm-09-00644-f002:**
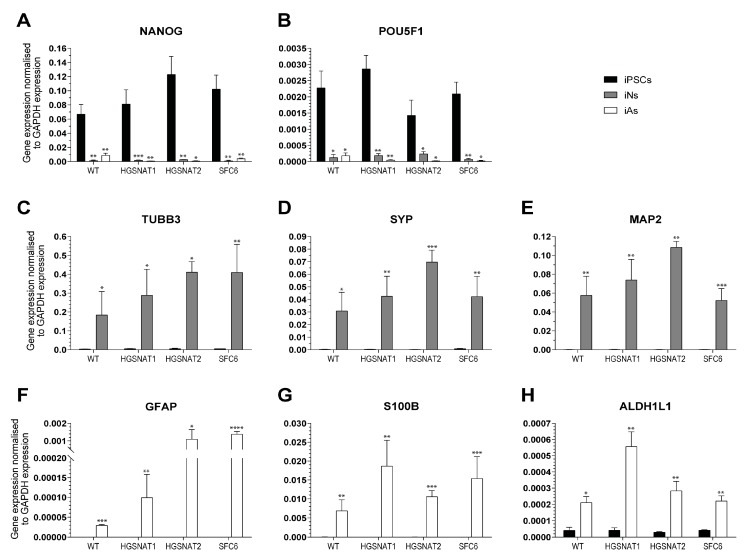
Cell-type-specific marker expression measured by qPCR in iPSCs (black), iNs (grey), and iAs (white) from all lines: WT, HGSNAT1, HGSNAT2, and SFC6. NANOG (**A**) and POU51F1 (**B**) are iPSC-associated markers; TUBB3 (**C**), SYP (**D**), and MAP2 (**E**) are neuronal-specific markers; and GFAP (**F**), S100B (**G**), and ALDH1L1 (**H**) are characteristic markers for astrocytes. Gene expression is normalized to GAPDH expression. Data are shown as the mean ± s.e.m. from three independent experiments with two technical replicates. **** *p* value < 0.0001, *** *p* value < 0.001, ** *p* value < 0.01, * *p* value < 0.05, ratio paired t-test comparing iPSC vs. iN or iAs.

**Figure 3 jcm-09-00644-f003:**
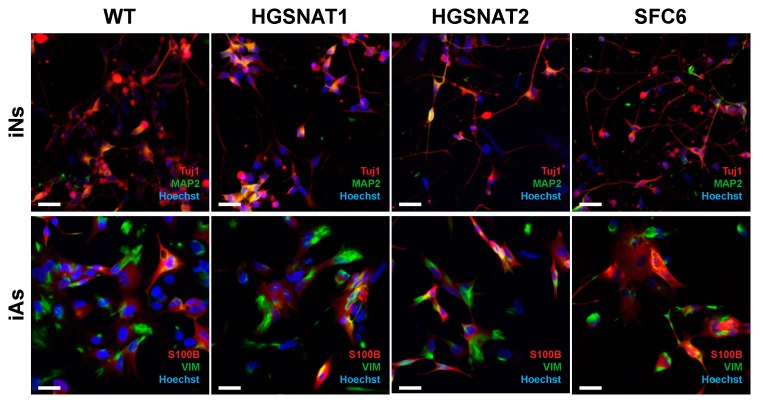
Representative images of differentiated iNs from all iPSC lines (WT, HGSNAT1, HGSNAT2, and SFC6) positive for Tuj1 (red) and MAP2 (green) and iAs from all iPSC lines positive for S100B (red) and VIM (green). Nuclei were detected with Hoechst (blue). Scale bar = 25 µm.

**Figure 4 jcm-09-00644-f004:**
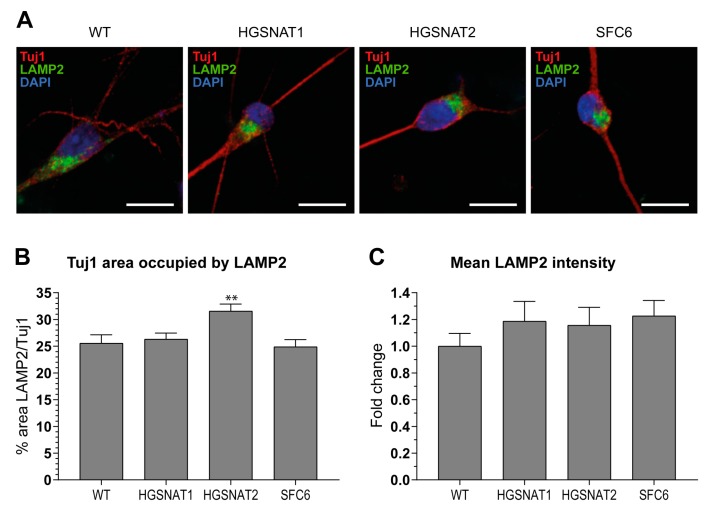
LAMP2 analysis in iNs. (**A**) Representative images of iNs after 10 days of differentiation positive for LAMP2 (green) and Tuj1 (red). DAPI was used to detect nuclei (blue). Scale bar = 10 µm. (**B**) Quantification of the lysosomal content shown as LAMP2+ coverage of the Tuj1+ neuronal soma. (**C**) Fold change in the mean LAMP2 dot intensity in disease iNs normalized to mean intensity of LAMP2 in WT-iNs. Data are shown as the mean ± s.e.m. from three independent experiments with five technical replicates. ** *p* value < 0.01, ordinary one-way ANOVA corrected by Dunnett post hoc test, comparing all groups to WT.

**Figure 5 jcm-09-00644-f005:**
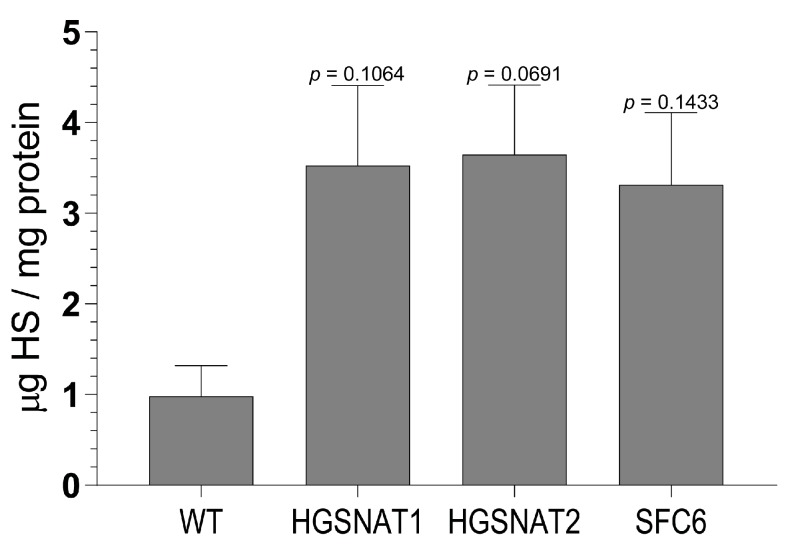
HS quantification by ultra-performance liquid chromatography–tandem mass spectrometer (UPLC-MS/MS) in iNs derived from all iPSC lines: WT, HGSNAT1, HGSNAT2, and SFC6. HS quantity is expressed in µg of HS in relation to mg of total protein. Data are shown as mean ± s.e.m. from three independent experiments with two technical replicates. One-way ANOVA corrected by Dunnett post hoc test, comparing all groups with the WT.

**Figure 6 jcm-09-00644-f006:**
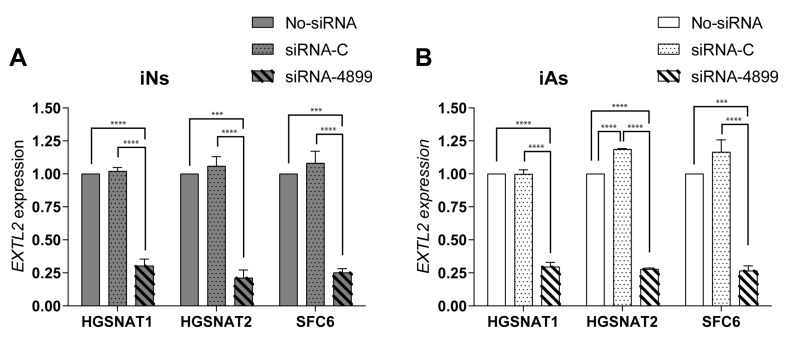
*EXTL2* mRNA expression after siRNA treatment in iNs (**A**) and iAs (**B**) derived from three different *HGSNAT*-mutated cell lines: HGSNAT1, HGSNAT2, and SFC6. Expression of *EXTL2* mRNA was normalized to GAPDH and relative to transfection with no-siRNA. Data are shown as mean ± s.e.m. from three independent experiments with two technical replicates. **** *p* value < 0.0001, *** *p* value < 0.001, ordinary one-way ANOVA corrected by Tukey post hoc test, comparing all three groups between each other within each cell line.

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
