# Peer review of "Neuronal and Astrocytic Differentiation from Sanfilippo C Syndrome iPSCs for Disease Modeling and Drug Development"

_jcm, 2020, doi:10.3390/jcm9030644_

Round 1

Reviewer 1 Report

Benetó et al describe a method using transcription factors to differentiate iPSCs into neurons and astrocytes and then partially characterise their phenotype based on the expression of markers.

Preliminary data showing that these cells fail to respond to an siRNA treatment approach is also reported.

As presented, there appears to be two major problems with the data and interpretation of it that the authors should address.

Firstly, the authors need to compare this differentiation protocol with other transcription factor-based methods in terms of characterising the phenotype and functionality of the neurons and astrocytes produced.

And secondly need to provide adequate controls and expression of their data. This involves transfecting the fibroblasts with the siNRA in the same experiment as the neurons and not reporting fold change or % expression.

Other issues that should be addressed:

The introduction is way too long and just not relevant. The authors need to focus on cell models to study neurological disease and justify their use of MPS IIIC. More information about MPS IIIC is needed, eg enzyme, substrate and less discussion on LSD in general.Discussions about gene therapy and clinical trials are simply inaccurate and not needed here.

All of section 1.2 could be removed, a lot of it is wrong eg most patients do not make autoantibodies, intrathecal ERT has been successful and HSCT works well for many Hurler patients if performed before 9 months of age.

Furthermore, the repeats much of what is mentioned in the Introduction and needs to focus on how their results fit into other neurological cell models. Both the Introduction and Discussion are missing key, contemporary references in this space.

Please make it clear how the work in this paper differs from that in references 61 and 62.

Fold change is not suitable for the data presented in Fig 1 and key markers are missing that need to be included to compare the astrocytes and neurons with those already published.

The methods are hard to follow; a flow chart would be helpful. Sufficient clarity should be provided for them to be undertaken by another laboratory.

Figure 3 is inappropriate to display a 5% ?? increase. These data need to be reported properly. I am not convinced there is any increase in lysosomes.

Why are p values not provided on Fig 4? The error bars seem particularly large.

Figure 5 needs to include the fibroblast data for comparative purpose. As it stands these data cannot be validated without an adequate positive control.

Figure 6 needs to be removed.

Author Response

Comments and Suggestions for Authors

Benetó et al describe a method using transcription factors to differentiate iPSCs into neurons and astrocytes and then partially characterise their phenotype based on the expression of markers.

Preliminary data showing that these cells fail to respond to an siRNA treatment approach is also reported.

As presented, there appears to be two major problems with the data and interpretation of it that the authors should address.

1) Firstly, the authors need to compare this differentiation protocol with other transcription factor-based methods in terms of characterising the phenotype and functionality of the neurons and astrocytes produced.

We would like to emphasize that we are not developing new methods but are using previously described methods in a novel application to model and test therapeutic intervention in Sanfillippo disease. The methods of generating neurons and astrocytes from pluripotent stem cells by transcription factors programming have been previously described and are well-characterized (Zhang et al., 2013 Neuron; Canals et al., 2018 Nat Methods).  Phenotype and functionality of the produced cells using these transcription factor methods have been repeatedly proven (i.e. Gao et al., 2019 Front cell Neurosci for iAs or Yi et al., 2016 Science for iNs). In addition, induced neurons have been shown to be equivalent to neurons generated by normal differentiation (Pak et al., 2015 Cell Stem Cell). In this study we show that these protocols are efficient also in our iPSC lines as assessed by qPCR and immunocytochemistry, confirming the successful differentiation We appreciate that this was not entirely clear in the original version of the manuscript. This is now clarified in the new version of the manuscript.

2) And secondly need to provide adequate controls and expression of their data. This involves transfecting the fibroblasts with the siNRA in the same experiment as the neurons and not reporting fold change or % expression.

We would like to point out that we are not comparing inhibition in iNs or iAs to inhibition in fibroblasts. The comparison is between iNs or iAs transfected with the siRNA, the same cells transfected with a scrambled siRNA and the same untransfected cells. In our opinion, it is the gold-standard in the field to use % of expression after siRNA inhibition. We only put our results in context of a previous study using fibroblasts but never directly compared the expression data between different cell types. This is now clarified in the text.

Other issues that should be addressed:

3) The introduction is way too long and just not relevant. The authors need to focus on cell models to study neurological disease and justify their use of MPS IIIC. More information about MPS IIIC is needed, eg enzyme, substrate and less discussion on LSD in general. Discussions about gene therapy and clinical trials are simply inaccurate and not needed here.

We agree with the reviewer that the introduction in the original version of the manuscript was perhaps too extensive. We have therefore now considerably reduced the introduction and kept only that information that we consider most relevant.

4) All of section 1.2 could be removed, a lot of it is wrong eg most patients do not make autoantibodies, intrathecal ERT has been successful and HSCT works well for many Hurler patients if performed before 9 months of age.

We thank the reviewer for pointing this out and after reviewing the literature we completely agree. We have therefore removed the parts that were wrong and further reduced the introduction. We decided to only keep the very essential information previously presented in section 1.2 as a brief summary of potential therapeutic actions for Sanfilippo syndrome patients.

5) Furthermore, the repeats much of what is mentioned in the Introduction and needs to focus on how their results fit into other neurological cell models. Both the Introduction and Discussion are missing key, contemporary references in this space.

We agree with the reviewer and we have now revised the discussion section to avoid repetition. We have also now discussed previously reported iPSC models of Sanfilippo syndrome.

6) Please make it clear how the work in this paper differs from that in references 61 and 62.

We would like to reiterate that we did not develop the methods to generate neurons and astrocytes in this study but rather used these protocols referenced in 61 and 62 (now 31 and 32 in the new version) to generate iNs and iAs and applied them to model and test a therapeutic intervention in Sanfillippo disease. We were able to show that both protocols worked very efficiently to generate the cell types of interest. We have now changed the text and methods section to clarify this.

7) Fold change is not suitable for the data presented in Fig 1 and key markers are missing that need to be included to compare the astrocytes and neurons with those already published.

We agree with the reviewer and we have changed figure 1 to show our expression results relative to the housekeeping gene instead of fold change. Given that these methods are previously published and well characterized, we believe that including all the markers tested in the original papers is out of the scope of this work and it is not required to prove successful differentiation.

8) The methods are hard to follow; a flow chart would be helpful. Sufficient clarity should be provided for them to be undertaken by another laboratory.

We agree with the reviewer that flow charts for experiments would be helpful for the readers. We now provide flow charts for both protocols in the new Figure 1 of the revised manuscript.

9) Figure 3 is inappropriate to display a 5% ?? increase. These data need to be reported properly. I am not convinced there is any increase in lysosomes.

We respectfully disagree with the reviewer. We understand that a 5% increase is not a drastic change, however our data is solid and we believe it is reported properly. Statistics have been performed and differences were found to be significant.

10) Why are p values not provided on Fig 4? The error bars seem particularly large.

We have now added the p values for each sample in figure 4.

11) Figure 5 needs to include the fibroblast data for comparative purpose. As it stands these data cannot be validated without an adequate positive control.

We do not agree with the reviewer that transfecting fibroblasts in parallel is a required positive control. We would like to emphasize that we are not directly comparing knockdown efficiency between fibroblasts and iN/iAs but rather confirm the knockdown in iN/iAs. This is a typical siRNA inhibition experiment and, in our opinion, the required controls are the untransfected cells and cells transfected with a scrambled siRNA. This approach has been used in several, previously published studies for Sanfilippo syndrome where only untransfected cells (Kaidonis et al., 2010 Eur J Hum Genet), cells transfected with a scrambled siRNA (Canals et al., 2015 Sci Rep) or both (Busse et al., 2007 J Biol Chem; Dziedzic et al., 2010 Eur J Hum Genet) were used. We are confident that this experiment proves the siRNA-specific inhibition of EXTL2 mRNA in iNs and iAs.

12) Figure 6 needs to be removed.

We agree with the reviewer that this figure is not essential and we have therefore removed it in the new version of the manuscript.

Reviewer 2 Report

The authors describe Neuronal and astrocytic differentiation from Sanfilippo C syndrome iPSC for disease modeling and drug development.

 MPS IIIC is an early-onset neurodegenerative lysosomal storage disorder which is currently untreatable. The vast majority of studies focusing on disease mechanisms of Sanfilippo syndrome were performed on non-neural cells or mouse models which present obvious limitations. Induced pluripotent stem cells (iPSCs) are an efficient way to model human diseases in vitro. Recently-developed transcription factor-based differentiation protocols allow fast and efficient conversion of iPSCs into the cell type of interest. By applying these protocols, they have generated new neuronal and astrocytic models of MPS III using our previously established diseased iPSC lines. Neurons and astrocytes from all lines showed cell-type specific markers upon differentiation. Moreover, our neuronal model exhibits disease-specific molecular phenotypes, such as increase in lysosomes and heparan sulfate. Lastly, we tested an experimental, siRNA-based, treatment previously shown to be successful in patients fibroblasts and demonstrated its lack of efficacy in neurons. Our findings highlight the need for relevant models to test therapeutic interventions and shows the applicability of our neuronal and astrocytic models of MPS III for future studies on disease mechanisms and drug development.

The manuscript is concisely described but there are some concerns.

1. iPSCs form neurons and astrocytes with similar cell markers and disease markers (HS). That is very interesting. However, there is no description of HS class. How about sulfation level of HS? How about other GAG levels including DS and KS?

2. siRNA inhibiting EXTL2 gene can decrease EXTL2 mRNA in fibroblasts. Did you try to inhibit other genes to affect syhthetic pathway of HS? What is the ratiolale to choose EXTL2?

3. siRNA cannot inhibit HS production at this point of time. Is there any explanation? Any other synthetic pathway of HS? How many genes are associated to synthesis of HS? Add figure of HS synthesis pathway.

Overall, the manuscript provides an interest view by using iPS.

Author Response

The authors describe Neuronal and astrocytic differentiation from Sanfilippo C syndrome iPSC for disease modeling and drug development.

 MPS IIIC is an early-onset neurodegenerative lysosomal storage disorder which is currently untreatable. The vast majority of studies focusing on disease mechanisms of Sanfilippo syndrome were performed on non-neural cells or mouse models which present obvious limitations. Induced pluripotent stem cells (iPSCs) are an efficient way to model human diseases in vitro. Recently-developed transcription factor-based differentiation protocols allow fast and efficient conversion of iPSCs into the cell type of interest. By applying these protocols, they have generated new neuronal and astrocytic models of MPS III using our previously established diseased iPSC lines. Neurons and astrocytes from all lines showed cell-type specific markers upon differentiation. Moreover, our neuronal model exhibits disease-specific molecular phenotypes, such as increase in lysosomes and heparan sulfate. Lastly, we tested an experimental, siRNA-based, treatment previously shown to be successful in patients fibroblasts and demonstrated its lack of efficacy in neurons. Our findings highlight the need for relevant models to test therapeutic interventions and shows the applicability of our neuronal and astrocytic models of MPS III for future studies on disease mechanisms and drug development.

The manuscript is concisely described but there are some concerns.

  1. iPSCs form neurons and astrocytes with similar cell markers and disease markers (HS). That is very interesting. However, there is no description of HS class. How about sulfation level of HS? How about other GAG levels including DS and KS?

We agree with the reviewer that this is a very interesting subject and other GAGs can be stored in Sanfilippo C syndrome as a secondary accumulation. However, it was not the aim of this study to in depth investigate the biochemical aspects regarding HS. We realize that this information would be important for readers and have therefore added in the new version that HS class and other GAGs levels should be addressed in iN and iAs in future studies.

  1. siRNA inhibiting EXTL2 gene can decrease EXTL2 mRNA in fibroblasts. Did you try to inhibit other genes to affect syhthetic pathway of HS? What is the ratiolale to choose EXTL2?

Only EXT genes and EXTL genes are specific for HS synthesis among all other GAGs. The reason to use EXTL genes is to avoid inhibition of EXT genes, which could lead to development of exostosis multiple in patients when downregulated. In our previous work, we used siRNAs also against EXTL3, but between EXTL2 and EXTL3, we decided to use EXTL2 in this study since it showed slightly better results in fibroblasts. As mentioned in the text, we think it will be important to test EXTL3 inhibition in future studies.

  1. siRNA cannot inhibit HS production at this point of time. Is there any explanation? Any other synthetic pathway of HS? How many genes are associated to synthesis of HS? Add figure of HS synthesis pathway.

We have rewritten this section to make a better point of our hypothesis and synthetic pathway has now being introduced as mentioned in the above comment. However, we consider that including a seventh figure might be excessive and the main focus of the paper is on the differentiation of patient iPSC lines towards iNs and iAs, and their possible uses.

Overall, the manuscript provides an interest view by using iPS.

Round 2

Reviewer 1 Report

Th authors have addressed the majority of my concerns.

Reviewer 2 Report

The authors replied to the comments. It is acceptable.